# Structural and Luminescence Properties of $(Gd_{1-x}Y_x)_2O_3$ Powders Doped with $Nd^{3+}$ Ions for Temperature Measurements

Vladimir Aseev [1,*], Anastasiia Babkina [1], Sergey Evstropiev [1] (ID), Natalia Kuzmenko [1], Irina Sevastianova [1], Denis Prokuratov [2] and Mikhail Khodasevich [3]

1   Faculty of Photonics, ITMO University, 197101 St. Petersburg, Russia
2   Department of Scientific and Technological Authentication of Works of Art, The State Hermitage Museum, 190000 St. Petersburg, Russia
3   Center "Diagnostic System", B. I. Stepanov Institute of Physics, 220072 Minsk, Belarus
*   Correspondence: aseev@oi.ifmo.ru

**Abstract:** Rare earth activated powders are widely regarded as promising candidates for optical thermometry due to their unique photoluminescence characteristics. The paper presents the structural and luminescent properties of crystalline powders of gadolinium and yttrium oxides $(Gd_{1-x}Y_x)_2O_3$, doped with $Nd^{3+}$ ions, synthesized by the liquid polymer-salt method. The addition of polyvinylpyrrolidone increases the homogeneity of the mixture and ensures high adhesion of the resulting powders. Scanning electron microscopy shows that powders are μm-sized aggregates, which consist of particles with several tens of nanometers in size. A smooth shift of the diffraction peaks of the powders occurs when Gd is replaced by Y without additional peaks. The successive decrease in the lattice constant of the powders from 10.816 to 10.607 Å confirms the existence of continuous solid solutions in the system. The Stark sublevels of the $^4F_{3/2} \rightarrow {}^4I_{9/2}$ fluorescent band are shifted to 4 nm when Gd is replaced by Y since the strength of the local field has a stronger effect on the inner F-shell of Nd ions in the case of Y. For thermometry, we chose the ratio of the fluorescence intensities between the Stark sublevels $^4F_{3/2(2)} \rightarrow {}^4I_{9/2(2)}$ and $^4F_{3/2(1)} \rightarrow {}^4I_{9/2(2)}$. The best obtained sensitivity is 0.22% $°C^{-1}$ for Nd-doped $GdYO_3$ powder in the range of 10–70 °C. This value of temperature sensitivity, together with radiation and excitation lying in the biological window, opens the possibility of using $Nd^{3+}$-doped $(Gd_{1-x}Y_x)_2O_3$ powders for real-time thermal probing of under tissue luminescence with sub-degree resolution.

**Keywords:** fluorescence thermometry; neodymium doped crystalline powder; relative thermal sensitivity; fluorescence intensity ratio; biological windows

## 1. Introduction

Temperature represents an important regulatory function in all biological systems controlling physiological and biochemical processes from the macroscale to the nanoscale [1]. In biomedicine, one of the most important tasks is the precise evaluation of local temperature inside a living organism (in vivo), for example, detecting inflammation or tumor. Temperature measurement using fluorescence is based on various physical principles. The most precise luminescent method is the temperature redistribution of energy over excited levels, leading to a redistribution of the fluorescence intensity in thermo-coupled bands [2,3].

Fluorescent thermal sensing and deep tissue imaging using nanomaterials emitting in biological transparency windows (the first window is from 700 to 980 nm, the second is from 1000 to 1350 nm, and the third is from 1550 to 1870 nm) are of great interest [4]. Visualization by radiation inside transparency windows is characterized by the possibility of deeper penetration into biological tissues and increased spatial resolution due to smaller absorption and light scattering compared to UV and visible radiation. In biomedical

applications, $Nd^{3+}$ is used as a dopant, which is excited by IR radiation and emits in the same spectral region [5]. Neodymium $^4F_{5/2} \rightarrow {}^4I_{9/2}$ fluorescence bands fall into the first biological transparency window, in which the main tissue constituents, such as water and hemoglobin, are transparent [6]. Moreover, neodymium can be excited by 800 nm radiation, which is preferable owing to the lower level of absorption and scattering by bio tissues in comparison with 980 nm [7].

The synthesis of rare-earth sesquioxides has generated significant interest due to their excellent chemical stability, good thermal conductivity and transparency to infrared radiation [8,9]. $Y_2O_3$-based ceramics have been used as efficient host matrix materials due to their high energy band gap (5.8 eV), broad transparency region, high melting point (2450 °C), and low thermal expansion coefficient [10]. Low phonon frequency, thermal and chemical stability, and low cytotoxicity arise as relevant features for the increasing interest in biological applications of $Gd_2O_3$ and $Y_2O_3$ [5,11].

Thus, in this article, we synthesized $(Gd_{1-x}Y_x)_2O_3$ sesquioxides crystalline powders doped with $Nd^{3+}$ ions and investigated their structural and spectral properties under the equimolar substitution of Gd for Y, as well as their temperature sensitivity, the accuracy of temperature determining, and their competitiveness with already known nanothermometers in this spectral range.

## 2. Materials and Methods

High-purity aqueous solutions of $Y(NO_3)_3 \times H_2O$, $Gd(NO_3)_3 \times 6H_2O$ and $Nd(NO_3)_3 \times 6H_2O$ were applied as precursors for $(Gd_{0.995-x}Y_xNd_{0.01})_2O_3$ phosphors synthesis, where x = 0; 0.245; 0.495; 0.745; 0.995. In the future, for simplification, we will not write the neodymium additive in the formula but imply it everywhere. Given volumes of these solutions, together with a solution of high molecular polyvinylpyrrolidone (PVP 10, average molecular weight 10,000, Sigma-Aldrich, St. Louis, MO, USA), were mixed with vigorous stirring for 30 min at room temperature. The addition of PVP should not only increase the homogeneity of the mixture but also ensure high adhesion of the resulting powders. After drying the solution at a temperature of 70 °C for 40 h, the homogeneous mixture was subjected to heat treatment in an electric muffle furnace at various temperatures: 600, 900, and 1100 °C for 2 h. The content of $Nd_2O_3$ was 1 wt.%.

X-ray diffraction patterns were obtained on a Rigaku Ultima IV (Rigaku Corporation, Tokyo, Japan) diffractometer with a copper anode and Bragg-Brentano focusing geometry. Experimental diffraction patterns were obtained at a wavelength of $\lambda CuK\alpha$ = 1.5418 Å in the range of 15–65° at a scanning rate of 0.5°/min. The picture was obtained using a tube voltage of 40 kV and a tube current of 40 mA. The crystal size was estimated by the Williamson-Hall and Halder-Wagner methods. The crystalline phase was then determined using the ICDD PDF-2 database. The morphology and chemical composition of the resulting powders were studied using a scanning electron microscope (SEM) Carl Zeiss EVO MA 25 (ZEISS Group, Jena, Germany) with an EDX detector X-MaxN 80 (Oxford instruments, High Wycombe, UK). The analysis was carried out at a voltage value of 20 kV and a counting time of 120 s for EDS analysis. The $^4F_{3/2} \rightarrow {}^4I_{9/2}$ luminescent band was recorded in the spectral range 850–980 nm on an AvaSpec-2048 spectrometer (Avantes, Apeldoorn, The Netherlands) and excited by a diode laser with a wavelength of 808 nm. When measuring the temperature, the powders were heated and cooled in a quartz cuvette using a Water Peltier System PCB 1500 (Perkin Elmer, Waltham, MA, USA) with a temperature programmer in the range of 10–71 °C with an accuracy of 0.1 °C.

## 3. Results and Discussion

### 3.1. XRD Studies

The X-ray Diffraction (XRD) results of powders synthesized at different temperatures showed the absence of an amorphous halo, and the calculation of the degree of crystallinity using the diffractometer software showed a value of about 72% (Figure 1). With an increase in temperature from 600 to 1100 °C during synthesis by low-temperature thermolysis, the

average grain size of the finished product increased from 6 to 43 nm (calculation error $\pm 0.5$ nm), and the degree of crystallinity increased from 31% to 72% (calculation error $\pm 5\%$), reaching a maximum at temperature synthesis at 900 °C within the error (inset in Figure 1). A further increase in temperature did not lead to an increase in the degree of crystallinity; therefore, for all samples described below, the annealing temperature was 900 °C. According to XRD data, "$Gd_1Y_1O_3$" powders do indeed contain cubic $GdYO_3$ crystals (PDF card No. 00-055-1053) with lattice period a = 10.706 $\pm$ 0.015 Å (C-type sesquioxide). An increase in the duration of the synthesis of $GdYO_3$ powders from an hour or more did not show an increase in the degree of crystallinity (Figure S1, in Supplementary Materials). For all samples, the single $GdYO_3$ cubic structure was obtained.

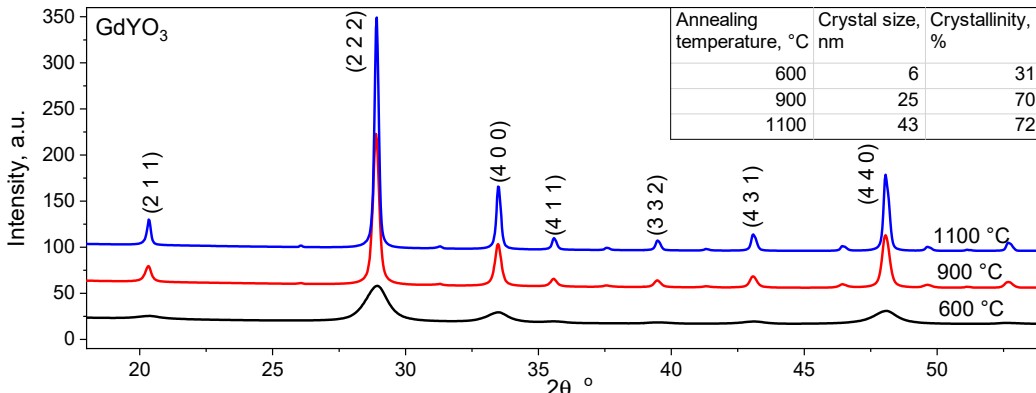

**Figure 1.** XRD patterns of $GdYO_3$ powders synthesized at different temperatures (inset: crystal size and crystallinity degree).

With equimolar substitution of gadolinium by yttrium ions, X-ray diffraction analysis showed that all synthesized samples had cubic symmetry. This was not surprising: sesquioxides of rare earth elements can have a hexagonal, monoclinic, and cubic structure depending on temperature [8]. At low temperatures, most oxides have a C-type cubic structure [12]. Depending on the calculation method and experimental technique, the lattice constant of cubic $Gd_2O_3$ varies from 10.812 to 10.843 Å [9,12,13], of bixbyite-type $Y_2O_3$ varies from 10.572 to 10.615 Å [14–16] at room temperature and normal pressure. In our case, the lattice constants were 10.816 and 10.607 Å for $Gd_2O_3$ and $Y_2O_3$, respectively.

Figure 2 shows a smooth shift of the diffraction peaks when Gd is replaced by Y without the appearance of any other peaks with a smooth decrease in the lattice constant. This is a strong indication that no other phase has formed. Thus, the presented system of oxides is a system of continuous solid solutions [17]: that is, when gadolinium is replaced by yttrium, no additional crystalline phases appear in the samples, but there is a smooth change in the parameters of the crystal lattice structure, which is shown in Table 1. The determined lattice parameters are in a linear relationship with the composition of the sample, i.e., follow Vegard's law.

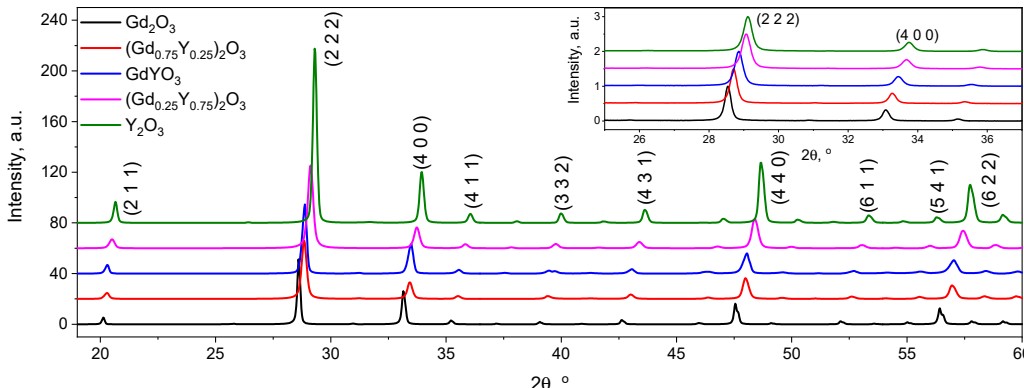

**Figure 2.** XRD patterns of $(Gd_{1-x}Y_x)_2O_3$ powders, where x = 0; 0.25; 0.5; 0.75, 1; the inset contains the most intense peaks on a larger scale.

**Table 1.** Crystal characteristics of $(Gd_{1-x}Y_x)_2O_3$ powders.

| Sample Composition | Crystal Phase/Card No. ICDD PDF-2 | Lattice Constant Calculated, Å | Cristal's Size, nm | Crystal System |
|---|---|---|---|---|
| $Gd_{1.98}Nd_{0.02}O_3$ | $Gd_2O_3/00\text{-}012\text{-}0797$ | 10.816 | 35 ± 1 | |
| $Gd_{1.49}Y_{0.49}Nd_{0.02}O_3$ | $Gd_6Y_4O_{15}/00\text{-}055\text{-}1055$ | 10.733 | 28 ± 3 | |
| $Gd_{0.99}Y_{0.99}Nd_{0.02}O_3$ | $GdYO_3/00\text{-}055\text{-}1053$ | 10.702 | 28 ± 3 | Cubic Ia-3 |
| $Gd_{0.49}Y_{1.49}Nd_{0.02}O_3$ | $GdY_9O_{15}/00\text{-}055\text{-}1049$ | 10.625 | 24 ± 3 | |
| $Y_{1.98}Nd_{0.02}O_3$ | $Y_2O_3/01\text{-}071\text{-}5970$ | 10.607 | 23 ± 3 | |

*3.2. Morphology Studies*

In order to observe the morphology of $(Gd_{1-x}Y_x)_2O_3$ powder, scanning electron microscopy was performed. The obtained photographs are shown in Figure 3. Powders consisted of μm-sized aggregates, which consist of particles with several tens of nanometers in size (Table 1). This morphology was due to the presence of PVP in the precursor solution at the stage of crystal powder synthesis. Due to the release of a large amount of gas formed from PVP during the annealing of solutions, the resulting oxide nanosized particles were nucleated in the form of an aggregated powder.

Chemical analysis of powders was carried out at several points for each composition and averaged. The values presented in Table 2 show that the actual composition of the synthesized powders within 2 wt.% coincided with the calculated composition. The maximum deviations were observed for yttrium. The mass loss of high-water yttrium nitrate used for synthesis during its thermal decomposition into water and yttrium oxide is, on average, 70%, while the decomposition reaction proceeds completely only up to 600 °C [18]. The mass loss of high-water gadolinium nitrate is 58%, and the decomposition reaction into water and gadolinium oxide proceeds completely only up to 730 °C [19,20]. Since, during the synthesis of powders, the initial reagents were weighed without taking into account the different volatility of the components, the yttrium content in the final product can be underestimated, which is shown in Table 2. Despite the simplicity of the synthesis method, the distribution of chemical elements in the composition of the final powder was uniform, as can be seen in Figure S2, and the deviation of the chemical composition of different points from the average was within 0.5%.

**Table 2.** The chemical composition of $(Gd_{1-x}Y_x)_2O_3$ crystalline powders (wt.%).

| Atom | $Gd_2O_3$ | | $Gd_{1.5}Y_{0.5}O_3$ | | $Gd_1Y_1O_3$ | | $Gd_{0.5}Y_{1.5}O_3$ | | $Y_2O_3$ | |
|------|------|----------|------|----------|------|----------|------|----------|------|----------|
| | Calc | Analysis | Calc | Analysis | Calc | Analysis | Calc | Analysis | Calc | Analysis |
| O | 13.25 | 13.25 | 14.61 | 14.45 | 16.29 | 15.90 | 18.41 | 18.49 | 21.15 | 21.13 |
| Gd | 85.95 | 85.74 | 71.11 | 72.83 | 52.85 | 57.43 | 29.85 | 29.13 | 0 | 0 |
| Y | 0 | 0 | 13.4 | 11.51 | 29.88 | 25.53 | 50.63 | 51.41 | 77.57 | 77.3 |
| Nd | 0.8 | 1.01 | 0.88 | 1.2 | 0.98 | 1.14 | 1.11 | 0.97 | 1.27 | 1.58 |

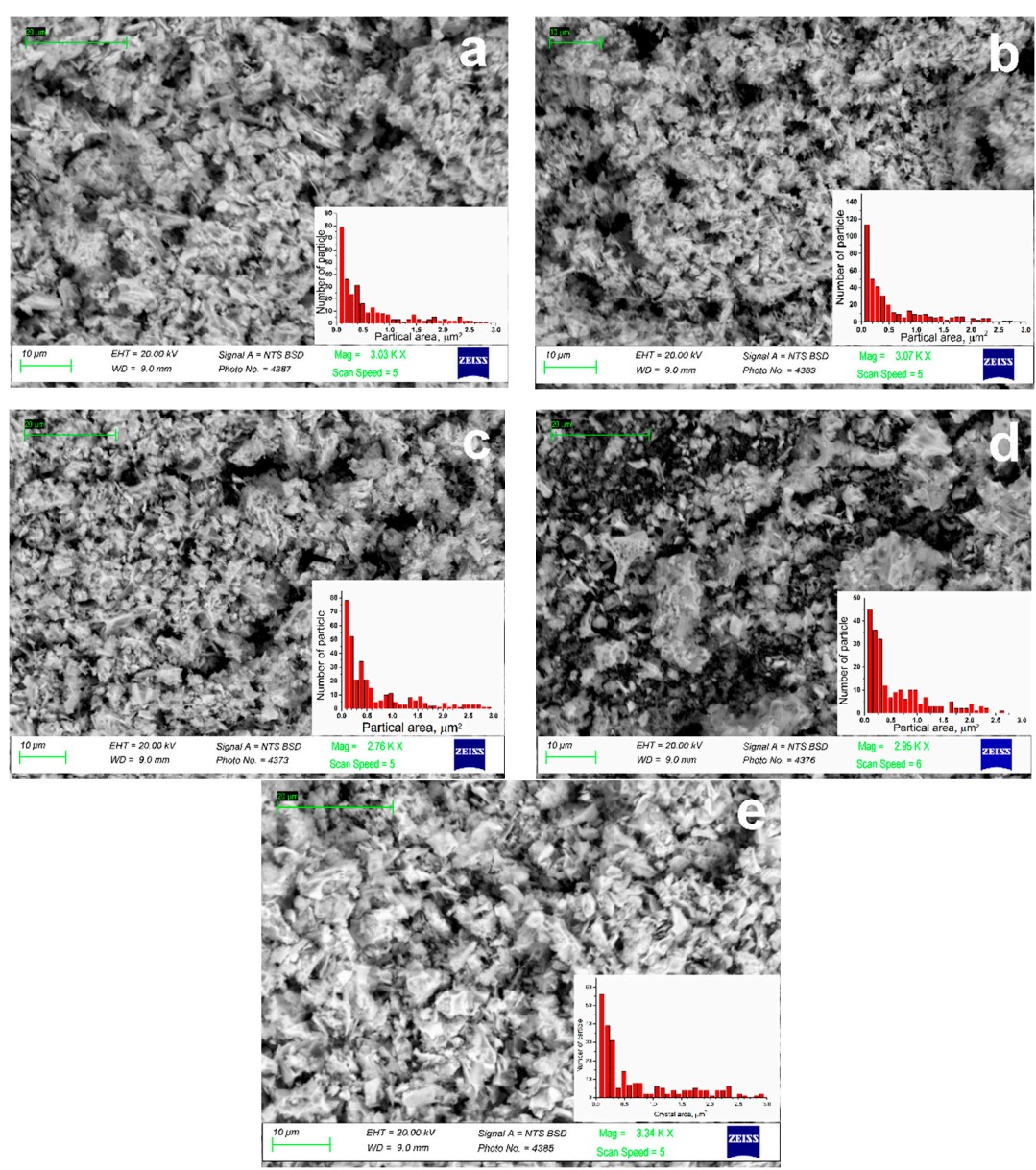

**Figure 3.** SEM images of $(Gd_{1-x}Y_x)_2O_3$ powders, where x = (**a**) 0; (**b**) 0.25; (**c**) 0.5; (**d**) 0.75; (**e**) 1.

*3.3. Spectral Properties*

The fluorescence spectrum contains three fluorescence bands corresponding to the $^2F_{3/2} \rightarrow {}^4I_{9/2}$, $^4I_{11/2}$ and $^4I_{13/2}$ transitions with maxima at 900, 1064 and 1350 nm, respectively (Figure 4).

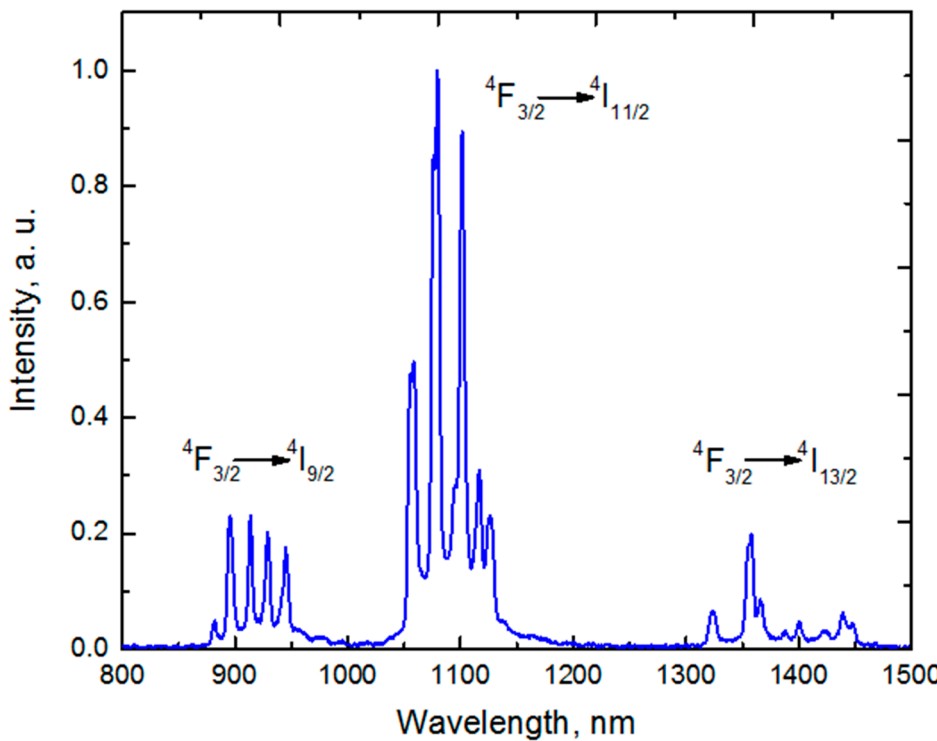

**Figure 4.** Fluorescence spectrum of $Gd_2O_3:Nd^{3+}$.

At least five emission peaks were clearly observed near 900 nm (Figure 5a). These lines can be associated with various transitions from $^4F_{3/2}$ to $^4I_{9/2}$ Stark sublevels [21].

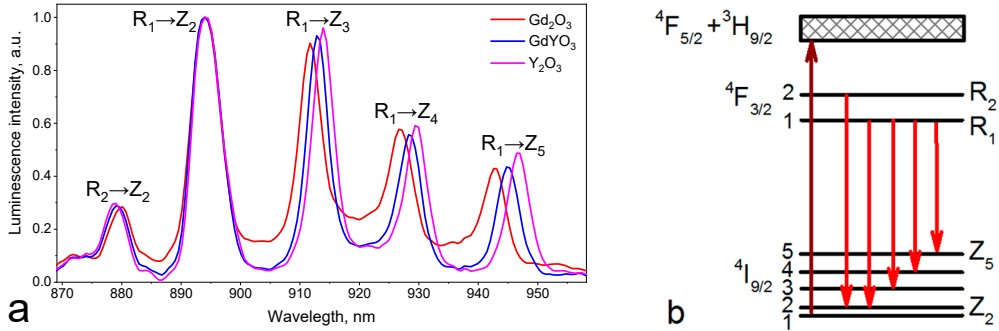

**Figure 5.** (**a**) Fluorescence spectra of $(Gd_{1-x}Y_x)_2O_3$ crystalline powders at $^4F_{3/2} \rightarrow {^4I_{9/2}}$ band; (**b**) corresponding transitions between Stark sublevels.

For the $RE^{3+}$ ions in the $C_2$ positions in $Y_2O_3$, "forced" electric dipole transitions between the Stark energy levels of the $^{2S+1}L_J$ $Nd^{3+}$ ($4f^3$) multiplet manifolds are allowed so that the $J + 1/2$ Stark levels expected in each manifold can be identified. By analyzing absorption and fluorescence spectra [22]. Comparing the energy difference between different Stark sublevels, we can conclude that the emission line centered at 879–880 nm was due to the transition from the higher-energy Stark sublevel $^4F_{3/2(2)}$ to $^4I_{9/2(2)}$, and the emission in the region of 943–947 nm was associated with the transition from the lower energy Stark sublevel $^4F_{3/2(1)}$ to $^4I_{9/2(5)}$ (Figure 5b) [23]. The band at 894 nm, corresponding to the $^4F_{3/2(1)} \rightarrow {^4I_{9/2(2)}}$ transition, remained immobile as the composition changed; it corresponded to the transition between thermally coupled levels and was taken into account when calculating the temperature sensitivity in what follows. The remaining lines were formed by the overlap of transitions from both $^4F_{3/2}$ Stark sublevels to different $^4I_{9/2}$ Stark sublevels [24]. It can be seen from Figure 5a that the Stark sublevels shifted as the composition changed from $Gd_2O_3$ to $Y_2O_3$, with the largest shift by 4 nm being observed

for the $^4F_{3/2(1)} \rightarrow {}^4I_{9/2(5)}$ transition. The shift value is presented in Table 3. The fluorescence spectrum of $Nd^{3+}$ ions in the region of the $^4F_{3/2} \rightarrow {}^4I_{11/2}$ transition (Figure S4) contained nine known transitions between the Stark sublevels [22] in the case of powder synthesis at 1100 °C.

**Table 3.** Influence of $(Gd_{1-x}Y_x)_2O_3$ composition on the luminescent transitions' location and crystal lattice parameters.

| Characteristics | $(Gd_{1-x}Y_x)_2O_3$ Nanopowder Composition | | | | |
|---|---|---|---|---|---|
| | x = 0 | x = 0.25 | x = 0.5 | x = 0.75 | x = 1 |
| $^4F_{3/2(1)} \rightarrow {}^4I_{9/2(3)}$ band location, nm ($\pm$0.5) | 911.7 | 912.5 | 913.1 | 913.8 | 914.5 |
| $^4F_{3/2(1)} \rightarrow {}^4I_{9/2(4)}$ band location, nm ($\pm$0.5) | 927.1 | 928.2 | 928.8 | 929.1 | 929.5 |
| $^4F_{3/2(1)} \rightarrow {}^4I_{9/2(5)}$ band location, nm ($\pm$0.5) | 943 | 944 | 945 | 946 | 947 |
| Energy gap between $R_1$ and $R_2$ sublevels, cm$^{-1}$ ($\pm$5) | 177 | 183 | 186 | 190 | 194 |
| Lattice constant, Å ($\pm$0.015) | 10.816 | 10.733 | 10.702 | 10.625 | 10.607 |

The shift occurred due to a change in the composition: when the environment of neodymium ions changes, the energy of the Stark sublevels changes depending on the composition. The wavelength of the fluorescence peak depends on the energy gap between the upper and lower levels. The local field strength affects the inner F-shell of neodymium ions in the case of substitution of Y for Gd since the size of the unit cell of $Gd_2O_3$, and $Y_2O_3$ is different (Table 3).

Thus, the Stark fluorescence sublevels shifted to longer wavelengths when Gd was replaced by Y. The effect of the unit cell size on the spectral properties of $Nd^{3+}$ can also be observed in the absorption spectra (Figure S3). According to Table 3, the dependence of the energy gap between the $R_1$ and $R_2$ sublevels of the $^4F_{3/2}$ level on the lattice constant can be approximated by a linear function (with an error of 5%). The principle of operation of thermoluminescent sensors is based on the redistribution of populations between two thermally coupled levels separated by an energy gap. The efficiency of the temperature sensor depends on the size of this gap since such thermometers suffer from low sensitivity due to the limitation of a small energy gap [7]. The standard gap between the $R_1$ and $R_2$ sublevels of the $^4F_{3/2}$ $Nd^{3+}$ level for $Y_2O_3$ is about 200 cm$^{-1}$ [24]. In our case, the maximum energy gap between adjacent Stark sublevels within the $^4F_{3/2}$ level was 195 cm$^{-1}$.

*3.4. Thermometry Studies*

Excitation near 800 nm promotes the $^4I_{9/2} \rightarrow {}^2H_{9/2} + {}^4F_{5/2}$ $Nd^{3+}$ transition [24]. To study the characteristics of the obtained samples as optical nanothermometers in biological systems, the emission spectra of samples doped with 1 wt.% $Nd^{3+}$ ions, upon excitation by a cw diode laser with a wavelength of 808 nm, were recorded, and the temperature of the sample varied in the range of 10–70 °C (Figure 5a), although the range of physiological temperatures is 30–60 °C. The change in temperature in the fluorescence spectra of other powders can be seen in Figure S5. An intensity redistribution of the fluorescence bands can be observed if the activator ion has two thermally coupled levels. The populations of such levels obey Boltzmann's law (Formula (1)):

$$\frac{N(R_1)}{N(R_2)} = \frac{g_{R_1}}{g_{R_2}}e^{(-\Delta E/kT)}, \tag{1}$$

where $N(R_i)$—the population of each of the thermally coupled levels; $g_{R_i}$—level degeneracy; $\Delta E$—is the energy gap between the levels, determined from the absorption or luminescence spectra; $k$—Boltzmann's constant.

That is, at temperatures above absolute zero, thermal energy can be enough to overcome the gap between thermally coupled levels, which leads to a redistribution of the population over the levels. This, in turn, leads to a change in the fluorescence bands' intensity corresponding to transitions from thermally coupled levels. After the establishment of a constant population at the levels during operation in a continuous mode, the ratio of the fluorescence bands corresponding to transitions from temperature-dependent levels can be related to temperature as follows:

$$R = \frac{I_{R_1}}{I_{R_2}} = Ce^{(-\Delta E/kT)},$$ (2)

where $I_{R_1}$ and $I_{R_2}$—integrated luminescence intensities of transitions from thermally coupled levels. The constant $C$ is defined as:

$$C = \frac{g_{R_1}\sigma_{R_1}\omega_{R_1}}{g_{R_2}\sigma_{R_2}\omega_{R_2}},$$ (3)

where $g$, $\sigma$, $\omega$—degeneracy, radiation cross section, angular frequency of fluorescent transitions from the corresponding thermally coupled levels to the main one.

During the study, we measured the fluorescence spectra of the neodymium ion, for which the role of thermally coupled levels is played by the Stark sublevels of the $^4F_{3/2}$ level—$R_1$ and $R_2$. Transitions are made to the $Z_2$ sublevel of the main $^4I_{9/2}$ level. As a result of the thermal population of the upper $R_2$ sublevel from the lower $R_1$ sublevel, the intensity of the band at about 880 nm increased relatively to the band at 894 nm with increasing temperature [22], i.e., comparing the integral intensities with each other these individual bands can provide temperature data.

The ratio of fluorescence intensities between $^4F_{3/2(2)} \rightarrow {}^4I_{9/2(2)}$ and $^4F_{3/2(1)} \rightarrow {}^4I_{9/2(2)}$ (hereinafter R) (Figure 6c,d) was chosen for thermometry since they have temperature dependence in the biological range due to the size of the energy gap between the Stark sublevels. Figure 6 shows the obtained fluorescence spectra, as well as the obtained fluorescence intensity coefficients and their approximation by an exponential function. The approximation parameters were the energy gap—a fixed value for each Y/Gd ratio and the coefficient C (Formula (3)). Parameter C was chosen programmatically to ensure maximum convergence [25].

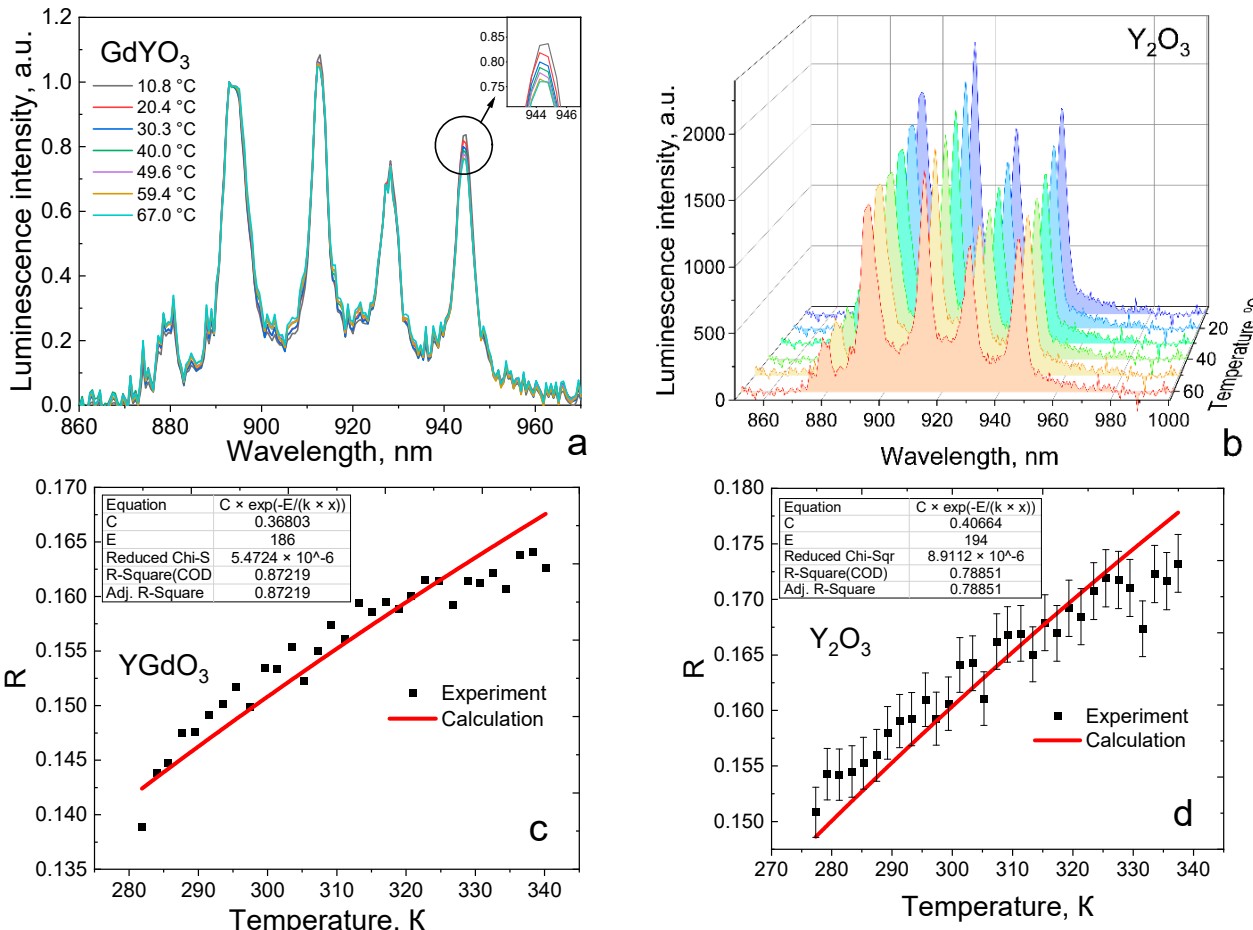

**Figure 6.** Fluorescence spectra' temperature dependence of Nd-doped GdYO$_3$ (**a**) and Y$_2$O$_3$ (**b**); Luminescence intensity ratio R as a function of temperature for GdYO$_3$ (**c**) and Y$_2$O$_3$ (**d**) powders. Red lines correspond to the fitting by an exponential function.

The approximation accuracy (adjusted R$^2$) lay in the range of 0.7 to 0.87. The incomplete correspondence between the experimental and theoretical dependences is due to the imperfection of the theoretical model. That is, that Equation (2) does not consider such phenomena as ion-ion interactions with temperature changes, an increase in temperature fluctuations, as well as various subtle effects. The maximum discrepancies are observed at the ends of the dependences since the contribution of temperature effects not taken into account in the theoretical model increases. That is, for the practical application of these crystalline systems as temperature sensors, it is necessary to use additional mathematical tools, such as regression analysis [26].

Figure S7 shows a graph of the dependence of the fluorescence intensity coefficient on temperature for different x from 0 to 1 in the (Gd$_x$Y$_{1-x}$)$_2$O$_3$ system. If we consider the entire temperature range in which the measurements were made, the coefficient R has maximum values for Y$_2$O$_3$:Nd (values lie in the range of 0.150–0.175) and minimum values for Gd$_2$O$_3$:Nd. The Gd$_{1.5}$Y$_{0.5}$O$_3$ sample did not fit into the general dependence, and its intensity coefficient was close in its values to that of yttrium oxide. This may be because at such a Gd/Y ratio, nonequivalent neodymium substitution sites appear and or the structure of the unit cell is distorted.

Absolute thermal sensitivity expresses the suitability of a fluorescent thermometer for measuring temperature. However, to compare the thermal performance of different

nanothermometers, regardless of their nature, it is useful to estimate the relative thermal sensitivity S, which can be expressed as follows [24,27]:

$$S = \frac{1}{R}\frac{dR}{dT} \times 100\%, \tag{4}$$

where $R$ is the mean size of luminescence intensity ratio, $dR$ express changes of luminescence intensity ratio with $dT$ changes of temperature. Table 4 shows the relative thermal sensitivity of all samples for $R$ fluorescence intensity ratios mean for the whole temperature region from 10 to 70 °C. The sensitivity of the intensity's ratio of the $R_2 \rightarrow Z_2$ to $R_1 \rightarrow Z_2$ transitions varies from 0.18 to 0.22, increasing from pure gadolinium oxide to GdYO$_3$. Table 4 shows that the sensitivity value for Gd$_2$O$_3$ doped with neodymium from [11] stands out from a number of other values. However, such a high sensitivity can be explained by the fact that the sensitivity was determined by comparing the fluorescence intensities of transitions from different excited levels and not between Stark sublevels. So, if we compare the sensitivity in terms of the ratio of fluorescence intensities between Stark transitions, then it lies in the range from 0.31 to 0.11% °C$^{-1}$. The maximum sensitivity we obtained was 0.22% °C$^{-1}$, which is a good result for this calculation method. To increase the sensitivity, it is possible to improve the mathematical processing of the obtained results, for example, using the partial least squares method [28]. The choice of spectral variables by the original method of searching for a combination moving window [26] in the multivariate model of partial least squares made it possible to reduce the mean square error of the temperature calibration of yttrium gadolinium oxide to 0.8 °C [29]. This result allows us to conclude that the proposed neodymium-doped crystalline powders and multivariate methods can be used to localize areas with febrile temperatures for biological and medical purposes.

**Table 4.** Comparison of Nd$^{3+}$ doped powders as luminescent thermometers.

| Host | Excitation Wavelength, nm | Transitions Used | Temperature Range, °C | Sensitivity S, % °C$^{-1}$ | Ref. |
|---|---|---|---|---|---|
| Gd$_2$O$_3$ | 808 | $R_2 \rightarrow Z_2$ to $R_1 \rightarrow Z_2$ | 10–70 | 0.18 | This work |
| Gd$_6$Y$_4$O$_{15}$ | 808 | $R_2 \rightarrow Z_2$ to $R_1 \rightarrow Z_2$ | 10–70 | 0.19 | |
| GdYO$_3$ | 808 | $R_2 \rightarrow Z_2$ to $R_1 \rightarrow Z_2$ | 10–70 | 0.22 | |
| GdY$_9$O$_{15}$ | 808 | $R_2 \rightarrow Z_2$ to $R_1 \rightarrow Z_2$ | 10–70 | 0.21 | |
| Y$_2$O$_3$ | 808 | $R_2 \rightarrow Z_2$ to $R_1 \rightarrow Z_2$ | 10–70 | 0.21 | |
| NaYF$_4$ | 830 | $I_{863}/I_{870}$ | 0–150 | 0.12 | [30] |
| NaYF$_4$ | 808 | $R_2 \rightarrow Z_1$ to $R_1 \rightarrow Z_1$ | 20–45 | 0.11 | [31] |
| YAG | 808 | $R_2 \rightarrow Z_5$ to $R_1 \rightarrow Z_5$ | 15–70 | 0.15 | [32] |
| YNbO$_4$ | 808 | $R_2 \rightarrow Z_2$ to $R_1 \rightarrow Z_2$ | 30–200 | 0.28 | [33] |
| LaF$_3$ | 808 | $I_{885}/I_{865}$ | 30–75 | 0.26 | [34] |
| LaF$_3$ | 808 | $I_{885}/I_{863}$ | 20–60 | 0.2 | [35] |
| YVO$_4$ | 808 | $R_1 \rightarrow Z_1$ to $R_2 \rightarrow Z_2$ | 25–60 | 0.19 | [36] |
| KGd(WO$_4$)$_2$ | 808 | $I_{895.8}/I_{883.8}$ | 20–65 | 0.12 | [37] |
| LiLaP$_4$O$_{12}$ | 808 | $R_2 \rightarrow Z_1$ to $R_1 \rightarrow Z_1$ | −190–20 | 0.31 | [38] |
| Gd$_2$O$_3$ | 580 | $^4F_{5/2} \rightarrow {}^4I_{9/2}$ to $^4F_{3/2} \rightarrow {}^4I_{9/2}$ | 15–50 | 1.75 | [11] |

## 4. Conclusions

Neodymium-doped crystalline powders of gadolinium and yttrium oxides $(Gd_{1-x}Y_x)_2O_3$ were synthesized by the liquid-polymer-salt method. The addition of PVP increased the homogeneity of the mixture and ensured high adhesion of the obtained powders. The synthesized crystals are 23–35 nm in size and are part of micrometer-sized agglomerates (up to 14 μm). The largest aggregates were observed in the $Gd_2O_3$ sample and the smallest in the $Y_2O_3$ sample. The $(Gd_{1-x}Y_x)_2O_3$ system was a system of continuous solid solutions showing a smooth shift in the diffraction peaks when replacing Gd with Y. The Stark sublevels of the $^4F_{3/2} \rightarrow {}^4I_{9/2}$ luminescent band shifted to 4 nm when Gd was replaced by Y since local fields had a stronger effect on the internal F-shell of Nd ions in the case of yttrium. Thermometers based on thermally coupled levels usually have low sensitivity due to the small energy gap limitation. The ratio of fluorescence intensities between $^4F_{3/2(2)} \rightarrow {}^4I_{9/2(2)}$ and $^4F_{3/2(1)} \rightarrow {}^4I_{9/2(2)}$ Stark sublevels was chosen for thermometry because of correspondence to the first biological window. The best obtained sensitivity was 0.22% $°C^{-1}$ for Nd-doped $GdYO_3$ powder in the range of 10–70 °C. These results may provide an effective way to improve thermometric performance and expand the application of fluorescent temperature measurement in various fields.

**Supplementary Materials:** The following supporting information can be downloaded at: https://www.mdpi.com/article/10.3390/ceramics5040084/s1. Figure S1. X-ray diffraction patterns of yttrium and gadolinium oxide $GdYO_3$ obtained during heat treatment at a temperature of 1100 °C for various durations. Figure S2. Distribution of chemical elements over the volume of $Gd_{0.5}Y_{1.5}O_3$ nanopowder obtained by SEM. Figure S3. Absorption spectra of $Nd^{3+}$ ions at $^4I_{9/2} \rightarrow {}^4F_{5/2}$, $^2H_{9/2}$ transitions in $(Gd_{1-x}Y_x)_2O_3$ samples, where x=0; 1; 2. Figure S4. Luminescence spectra of $Nd^{3+}$ ions at $^4F_{3/2} \rightarrow {}^4I_{11/2}$ transition in $Y_2O_3$ powder obtained during heat treatment at different temperatures for 1 hour. Figure S5. Luminescence spectra' temperature dependence of $Gd_2O_3$ (a) and $Gd_{1.5}Y_{0.5}O_3$ (b) samples. Figure S6. Luminescence intensity ratio R as a function of temperature for $Gd_2O_3$ (a) and $Gd_{1.5}Y_{0.5}O_3$ (b) samples. Red lines correspond to the fitting by exponential function. Figure S7. Luminescence intensity coefficients (R) for all samples from the series $(Gd_xY_{1-x})_2O_3$.

**Author Contributions:** Writing—original draft, V.A. and I.S.; methodology, V.A.; resources, V.A., A.B. and S.E.; investigation, S.E., I.S., N.K., D.P. and M.K.; supervision, V.A.; writing—review & editing, A.B. and N.K.; funding acquisition, A.B.; Formal analysis, N.K. All authors have read and agreed to the published version of the manuscript.

**Funding:** The reported study was funded by RFBR and BRFBR, project number 20-58-00054.

**Institutional Review Board Statement:** Not applicable.

**Informed Consent Statement:** Not applicable.

**Data Availability Statement:** Not applicable.

**Acknowledgments:** The research held at ITMO University was supported by the Priority 2030 Federal Academic Leadership Program.

**Conflicts of Interest:** The authors declare no conflict of interest.

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
