# Peer review of "Structural and Luminescence Properties of (Gd1−xYx)2O3 Powders Doped with Nd3+ Ions for Temperature Measurements"

_ceramics, doi:10.3390/ceramics5040084_

Round 1
Reviewer 2 Report
Review Manuscript ID ceramics-2015008. Structural and luminescence properties of (Gd1-xYx)2O3 nanopowders doped with Nd3+ ions for temperature measurements.
This work could be useful for people working in this field, and some contents are of interest. The manuscript could be published in Ceramics after some mayor revisions.
It would be desired to response the following comments and questions:
SEM and XED must be defined. Review all the acronyms along the manuscript are defined.
1. Introduction
Page 2. “The synthesis of rare-earth sesquioxides has generated significant interest due to their excellent chemical stability, good thermal conductivity and transparency to infrared radiation”. Reference must be included.
2. Methodology
What is the reason of 1 wt.% of Nd2O3? Did you think about the interest of other %?
3. Results and discussion
Pages 2 and 3. ”the degree of crystallinity increased from 31% to 72%, reaching a maximum at temperature synthesis at 900°C (inset in Figure 1)”. However, According to inset in Fig. 1, the maximum is not obtained at 900 ºC.
Page 3. What cubic structure, CS, BCC, FCC?
Page 3. “For all samples, the single GdYO3 cubic structure was obtained”. A picture of the cubic structure, showing the disposition of atoms, should be included.
Figure 2. Red and Pink represent the same sample but XRD spectra are different?
3.2. Morphology studies
Page 4. Authors must explain the calculation of granular particles size. The values presented (4 to 14 microns) should be related with the pictures of Figure 3.
What is the relevance of granular particles size for this research? A disccusion of this aspect could be explained.
3.3. Spectral properties
Page 6. “The luminescence spectrum contains three luminescence bands corresponding to the 2F3/2→4I9/2, 4I11/2 and 4I13/2 transitions with maxima at 900, 1064, and 1350 nm”. Where is thes spectrum showing these transitions? It must be reported.
Figure 4a. For R2®Z2 transition luminescence is shifted to longer wavelengths in contrast with other transition in which luminescence sublevels shifted to longer wavelengths. Authors must clarify this behavior.
Figure 4b. The absorption transition must be included in this figure.
These graphs show a high dispersion of the R vales. Can be this system considered as reliable for temperature sensor purposes?
Reviewer 3 Report
The author prepared (Gd1-xYx)2O3 by controlling the doping concentration of Y3+ ions. By replacing Gd3+ with Y3+, the lattice shrinks and part luminescence peaks shift. High optical temperature measurement sensitivity is realized by using the Stark sublevels 4F3/2(2)→ 4I9/2(2). The author needs to further explore the highlights in this research.
1. Page 2-3, the inset in Figure 1 shows that the degree of crystallinity reaches the maximum at 1100 ℃. However, the manuscript show that the degree of crystallinity increased from 31% to 72%, reaching a maximum at temperature synthesis at 900°C (inset in Figure 1).
2. Page 4, the sentence "In order to observe (Gd1-xYx)2O3 crystal’s morphology scanning electron microscopy was performed. " can be modified as "In order to observe the morphology of (Gd1-xYx)2O3 crystal, scanning electron microscopy was performed. "
3. Page 4, in Table 1, it should be "crystal" instead of " cristal".
4. Page 4, the SEM shows that powders consisted of granular particles ranging in size from 4 to 14 microns and did not contain larger particles or aggregates. However, the inset in Figure 1 and Table 1 show that the crystal size is tens of nanometers. Please check carefully whether the data is correct.
5. Please solve the inconsistency between "luminescence spectrum" and "fluorescence spectra".
6. In Table 1, how did the author distinguish the crystal phase of the sample, and on what basis?
7. The author needs to modify Figure 4 (b), otherwise it will be difficult for the reader to understand which energy level 4F3/2(1) and 4I9/2(5) refer to.
Round 2
Reviewer 1 Report
In my opinion, the manuscript is now acceptable for publication in the Ceramics journal.
Author Response
In my opinion, the manuscript is now acceptable for publication in the Ceramics journal.
The authors thank you for your comments that allowed us to improve the quality of our article.
Reviewer 2 Report
Second Review Manuscript ID ceramics-2015008. Structural and luminescence properties of (Gd1-xYx)2O3 nanopowders doped with Nd3+ ions for temperature measurements.
The authors have response some of the requested questions and comments. However, some questions have not been adequately revised. Also, the responses are mixed: some of the response to my questions are in the response to other reviews, and some of the responses to others reviews are in the response to my questions.
Before publication, it would be desired to response the following comments and questions:
Figure 2 has not been corrected in the final version. Sample labels are not coherent. Review exhaustively all the labels.
Figures 5c and 5d show a high dispersion of the R vales. Can be this system considered as reliable for temperature sensor purposes? Please, comment this result in the text of the manuscript. This is very important because this sample is studied for temperature measurements.
Author Response
Point 1. Figure 2 has not been corrected in the final version. Sample labels are not coherent. Review exhaustively all the labels.
Response 1 The corrected figure is added to the new revision of the article
Point 2. Figures 5c and 5d show a high dispersion of the R vales. Can be this system considered as reliable for temperature sensor purposes? Please, comment this result in the text of the manuscript. This is very important because this sample is studied for temperature measurements.
Response 2
The following clarification has been added to the new revision of the article:
The approximation accuracy (adjusted R2) lied in the range from 0.7 to 0.87. The incomplete correspondence between the experimental and theoretical dependences is due to the imperfection of the theoretical model. That is, that equation (2) does not consider such phenomena as ion-ion interactions with temperature changes, an increase in temperature fluctuations, as well as various subtle effects. The maximum discrepancies are observed at the ends of the dependences, since the contribution of temperature effects not taken into account in the theoretical model increases. That is, for the practical application of these crystalline systems as temperature sensors, it is necessary to use additional mathematical tools, such as regression analysis [29].